# Risk factors for catheter-associated bloodstream infection in hemodialysis patients: A meta-analysis

**Huajie Guo**[1☯]**, Ling Zhang**[2☯]**, Hua He**[3]*, **Lili Wang** [2]*

**1** Department of Urology, First Hospital of Shanxi Medical University, Taiyuan, Shanxi, China, **2** Department of Nursing, Shanxi Provincial People's Hospital, Taiyuan, Shanxi, China, **3** Intensive Care Unit, Shanxi Provincial People's Hospital, Taiyuan, Shanxi, China

☯ These authors contributed equally to this work.
\* hh18636119389@163.com (HH); 491543779@qq.com (LW)

**Data Availability Statement:** All relevant data are within the manuscript and its Supporting information files.

## Abstract

### Objective

This meta-analysis aimed to elucidate the risk factors contributing to catheter-associated bloodstream infection in hemodialysis patients.

### Methods

Comprehensive literature searches were conducted in both English and Chinese databases, which encompassed PubMed, Cochrane Library, Embase, CNKI, Wanfang Data, VIP Database and China Biomedical Literature Database. The search timeframe extended from each database's inception to March 8, 2023. Two independent researchers executed literature screening, data extraction, and quality assessment using the Newcastle-Ottawa Scale. Statistical analysis of the data was performed using RevMan 5.3 software, facilitating the identification of significant risk factors associated with catheter-related bloodstream infections in hemodialysis patients. This meta-analysis is registered with PROSPERO under the registration number CRD42023406223.

### Results

Forty-nine studies were incorporated into this meta-analysis, from which 22 risk factors were examined. Through the analysis, 17 risk factors exhibited statistical significance ($P < 0.05$): age (OR = 1.52, 95% CI [0.49, 4.68]), diabetes (OR = 2.52, 95% CI [1.95, 3.25]), kidney disease (OR = 3.45, 95% CI [1.71, 6.96]), history of catheter-associated infection (OR = 2.79, 95% CI [1.96, 3.98]), hypertension (OR = 1.43, 95% CI [1.08, 1.91]), dialysis duration (OR = 3.06, 95% CI [1.70, 5.50]), catheter placement site (OR = 1.91, 95%CI [1.35, 2.70]), catheter duration (OR = 2.06, 95% CI [1.17, 3.60]), number of catheterizations (OR = 4.22, 95% CI [3.32, 5.37]), catheter types (OR = 3.83, 95% CI [2.13, 6.87]), CD4+ cells (OR = 0.33, 95% CI [0.18, 0.63]), albumin (ALB, OR = 2.12, 95% CI [1.15, 3.91]), C-reactive protein (CRP, OR = 1.73, 95% CI [1.47, 2.03]), hemoglobin (Hb, OR = 1.48, 95% CI [0.54, 4.07]), procalcitonin

**Funding:** The authors received no specific funding for this work.

**Competing interests:** The authors have declared that no competing interests exist.

(PCT, OR = 1.05, 95% CI [1.03, 1.06]), inadequate hand hygiene (OR = 5.32, 95% CI [1.07, 26.37]), and APACHE II scores (OR = 2.41, 95% CI [1.33, 4.37]).

## Conclusion

This meta-analysis suggests that age, diabetes, kidney disease, history of catheter-associated infection, hypertension, dialysis duration, catheter placement site, catheter duration, number of catheterizations, catheter type, $CD4^+$ cells, albumin, C-reactive protein, hemoglobin, procalcitonin, inadequate hand hygiene, and APACHE II scores significantly influence the incidence of catheter-associated bloodstream infection in hemodialysis patients.

## 1 Introduction

As an irreversible kidney damage, chronic kidney disease (CKD) represents a significant health concern due to its potential to cause eventual renal failure. This condition is particularly prevalent in China, where it imposes a substantial socio-economic burden. Furthermore, the global health implications of CKD are profound. According to a recent meta-analysis, the prevalence of CKD in Chinese adults is approximately 13.2%, with an observed trend of increasing prevalence with advancing age [1]. In the clinical landscape, end-stage renal disease, a severe manifestation of CKD, is typically managed via kidney replacement therapy. This encompasses three main strategies: hemodialysis (HD), peritoneal dialysis, and kidney transplantation. Of these, HD is the most frequently employed, with over 90% of end-stage renal disease patients relying on it for management [2, 3].

The advent of advanced blood purification technology has necessitated the increased use of central venous catheters. However, this has consequently escalated the risk of catheter-related complications, including thrombosis, infection, stenosis, and malfunction [4]. Infections associated with catheter use in HD patients can be broadly classified into three categories: outlet infection, tunnel infection, and catheter-associated bloodstream infection (CRBSI). Although the incidence of outlet and tunnel infections has decreased due to improved aseptic techniques, CRBSI continues to be a significant challenge. If not treated promptly and effectively, CRBSI may lead to serious complications, including cardiovascular and cerebrovascular diseases, and increased mortality. Both patient-specific and treatment-related factors contribute to the risk of these complications [5].

Previous research exploring the risk factors for CRBSI in HD patients has identified two primary categories: patient-related factors and treatment-related factors. Patient-related factors include age, disease severity [6], and co-existing conditions such as breast cancer [7], lung cancer [8], hematologic malignancies [9], and diabetes [10]. Treatment-related factors encompass catheterization method and type, frequency and duration of catheterization, dialysis duration, nursing experience, and ward environment. However, due to variances in sample size, population, and research locations across studies, the identified risk factors exhibit some inconsistencies. In light of this, the present study aims to conduct a meta-analysis to explore the risk factors for CRBSI in HD patients, thereby seeking to reduce potential bias in research findings that may arise from inadequate sample sizes.

### 1.1 Registration

This meta-analysis is registered with PROSPERO under the registration number CRD42023406223.

## 2 Research design and methodology

### 2.1 Literature search strategy

To ensure comprehensive coverage, we conducted an exhaustive literature search across both English and Chinese databases. The English databases encompassed PubMed, Cochrane Library, and Embase. In contrast, the Chinese databases included China National Knowledge Infrastructure (CNKI), Wanfang Data, VIP Database, and China Biomedical Literature Database (CBM). To augment the robustness of our search, we manually examined the references of the selected articles to uncover any additional pertinent literature. The search timeframe spanned from the inception of each respective database through March 8, 2023.

### 2.2 Study selection criteria

The criteria for study inclusion comprised the following: (1) Only case-control or cohort studies were eligible; (2) The participants undergoing HD had to be at least 18 years old; (3) Studies should report influencing factors of CRBSI incidence in HD patients; (4) The control group should be devoid of any CRBSI instances; (5) The outcome measures should solely represent CRBSI incidence, excluding catheter-related local and tunnel infections, with results either presented or convertible into OR and 95% confidence interval (95% CI); (6) The articles had to be written in either Chinese or English.

The exclusion criteria were as follows: (1) Repetitive publications; (2) Studies with incompatible design; (3) Studies with ambiguous diagnostic criteria; (4) Studies with incomplete or absent basic data; (5) Studies where full text was unavailable; (6) Case reports, reviews, systematic reviews, expert opinions, animal studies, and conference papers; (7) Studies with erroneous statistical analysis methods.

### 2.3 Literature screening

All identified articles were uploaded to NoteExpress software. Two investigators independently carried out an initial screening of the titles and abstracts, strictly adhering to the predetermined selection criteria. This was succeeded by a meticulous full-text review of the preliminarily chosen articles. In the event of discrepancies over inclusion, a consensus was attained either through dialogue between the two investigators or consultation with a third investigator.

### 2.4 Data extraction

Two investigators independently gleaned the relevant information from the selected articles and cross-verified the data following extraction. Any disputes were settled through mutual agreement or by deferring to a third investigator. The extracted data primarily consisted of: (1) General information: article title, first author, year of publication, study period, country of study, study type, patient age, and sample sizes for both case and control groups, delineated by gender; (2) Outcome measures: related risk factors and corresponding OR values along with their 95% CI.

### 2.5 Evaluation of literature quality

Two trained researchers independently assessed the quality of the literature, with discrepancies resolved through discussion or adjudication by a third party. The Newcastle-Ottawa Scale (NOS), endorsed by various American health care and research institutions, served as the primary tool for assessing the quality of case-control and cohort studies [11]. The NOS encompasses three categories and eight items, including: selection of the study and control groups

(four items), comparability of the groups (one item), and evaluation of exposure or outcome (three items). The NOS scoring system assigns a maximum of nine points, with a score of six or higher indicating high literature quality [12].

## 2.6 Statistical analysis

Data were analyzed using Review Manager (RevMan) 5.3 software. Results were expressed as OR and 95% CI, with a *P*-value of less than 0.05 considered statistically significant. Heterogeneity of the study findings was evaluated using Q and $I^2$ tests. A *P*-value greater than 0.1 and an $I^2$ value less than 50% were indicative of statistical homogeneity across studies, in which case a fixed-effect model was employed for effect size synthesis. Conversely, should the *P*-value be less than or equal to 0.1 and the $I^2$ value be 50% or greater, denoting substantial heterogeneity, a random-effects model was utilized for analysis. Sensitivity analysis was conducted to assess the robustness of the study findings. If the results derived from the fixed and random-effects models were closely aligned, the findings were deemed stable. Conversely, significant divergence between the two models suggested potential instability in the results.

## 3 Results

### 3.1 Literature search and screening

The initial search of the English and Chinese electronic databases generated 1146 potential articles, with 812 sourced from Chinese databases and 334 from English databases. We initially utilized NoteExpress software for duplicates, followed by manual cross-checking to further refine the search. Through this process, 363 duplicate entries were successfully removed. Subsequent screening involved careful reading of titles and abstracts, resulting in the elimination of 528 irrelevant articles. Additionally, 61 pieces, including conference papers, literature reviews, systematic reviews, and meta-analyses, were excluded. We further excluded four articles due to the unavailability of full texts. After a comprehensive review of the full texts, another 141 articles were deemed unfit for inclusion. Ultimately, the final dataset consisted of 49 articles, 35 in Chinese and 14 in English, suitable for inclusion in the study. Fig 1 illustrates the literature screening process and its results.

### 3.2 General characteristics and quality evaluation of the included studies

The 49 articles included in the study consisted of 35 Chinese articles, all of which were case-control studies, and 14 English articles, of which 7 were case-control studies and 7 were cohort studies. The articles included in our study spanned a publication period from 2004 to 2023. Each of the 22 identified risk factors was investigated in at least two articles. In terms of quality assessment, the Newcastle-Ottawa Scale assessed the score between the two investigators, 44 of the 49 included studies scored between 7 and 9, indicative of high quality. The remaining five studies scored 6, suggesting medium quality. Table 1 provides a detailed overview of the general characteristics and quality evaluation of the included studies.

### 3.3 Results of meta-analysis

The meta-analysis was conducted on 22 risk factors identified for inclusion in this study, the findings of which are as follows:

(1) Age and its effect on CRBSI:

A total of thirteen studies in our analysis highlighted an association between age and the incidence of CRBSI in HD patients. Subgroup analysis was performed for different age stages

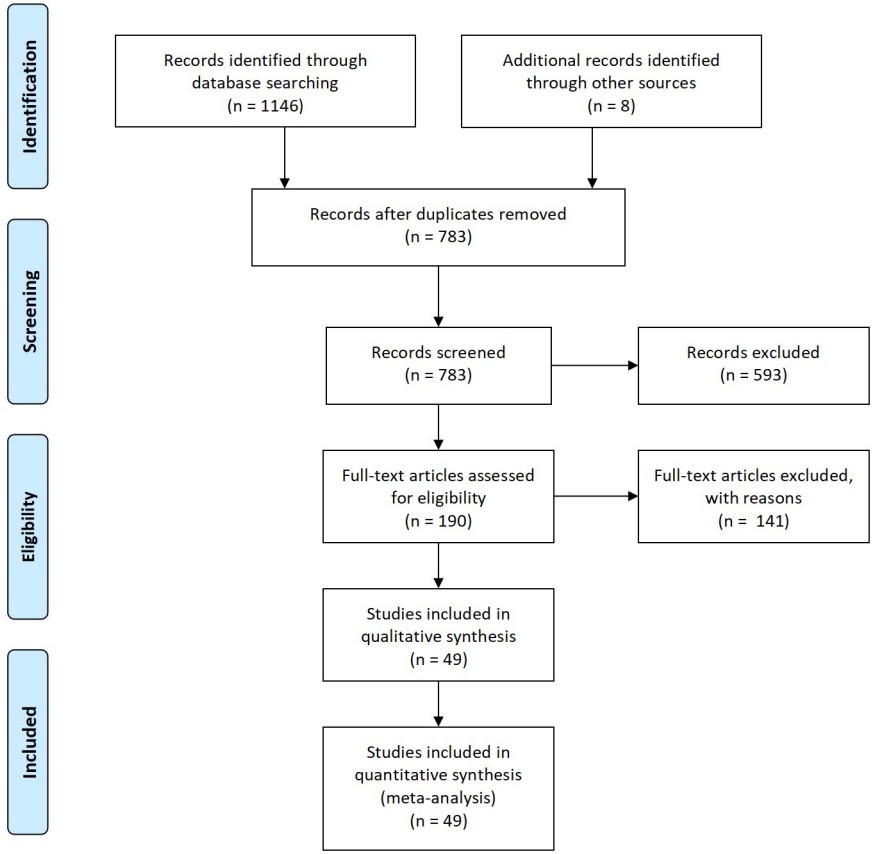

**Fig 1. Flow chart depicting the literature screening process.**

and they were divided into three different groups: $\geq$60 and $<$60 years old, $>$60 and $\leq$60 years old, $>$75 and $\leq$75 years old. The results of the study found no statistical heterogeneity between the first two groups of studies and the findings were statistically significant; In contrast, there was a statistically significant difference between studies in the third group ($P < 0.01$, $I^2 =$ 100%), with greater heterogeneity. A high level of statistical heterogeneity was observed across these studies ($I^2 = 96\%$, $P < 0.01$), necessitating the use of a random-effects model for the meta-analysis. The results indicated that advanced age is a significant risk factor for CRBSI (OR = 1.52, 95% CI [0.49, 4.68], $P = 0.47$). Fig 2 shows the forest plot for age.

(2) Gender and its effect on CRBSI:

Our analysis included two studies which reported a correlation between gender and the incidence of CRBSI in HD patients. Statistical heterogeneity among these studies was substantial ($I^2 = 84\%$, $P = 0.01$), therefore, a random-effects model was employed. The meta-analysis showed no statistically significant association between gender and CRBSI risk (OR = 1.02, 95% CI [0.20, 5.05], $P = 0.98$). Fig 3 shows the forest plot for gender.

(3) Diabetes mellitus and its effect on CRBSI:

Twenty-five studies in our analysis reported an association between diabetes mellitus and the incidence of CRBSI in HD patients. Given the significant heterogeneity among these studies ($I^2 = 69\%$, $P < 0.00001$), a random-effects model was utilized. The meta-analysis revealed

**Table 1. Study characteristics and quality evaluation of literature included in the meta-analysis.**

| Study | Publish year | Country | Investigation time | Research type | Age Case group/ Control group | Diagnostic criteria for CRBSI | Number of cases | Number of controls | Rask factors | Scores |
|---|---|---|---|---|---|---|---|---|---|---|
| An et al. [13] | 2022 | China | 2012.11–2017.11 | ① | 68–91 | a | 29 | 64 | 8,12 | 7 |
| Cao et al. [14] | 2019 | China | 2010.01–2018.06 | ① | —— | b | 78 | 236 | 2,5,12 | 7 |
| Cheng et al. [15] | 2019 | China | 2013.01–2018.12 | ① | 55.64±12.69 | c | 55 | 298 | 1,2,7,9,12,24 | 8 |
| Ding et al. [16] | 2021 | China | 2017.09–2019.09 | ① | 57.7±15.3 | a | 42 | 141 | 5,12,25 | 8 |
| Huang et al. [17] | 2017 | China | 2014.01–2015.12 | ① | 77.96±10.57 | c | 7 | 71 | 8,16 | 7 |
| Huang et al. [18] | 2017 | China | 2011–2015 | ① | 59.06±16.32 | b | 51 | 46 | 1,2,7,12,16 | 7 |
| Jiang et al. [19] | 2016 | China | 2012.01–2014.12 | ① | 57.50±11.3 | d | 60 | 1270 | 6,7,26 | 6 |
| Li et al. [20] | 2023 | China | 2019.01–2021.01 | ① | 58.23±7.11/62.29±8.35 | b | 20 | 160 | 2,7 | 7 |
| Li et al. [21] | 2021 | China | 2015.03–2020.03 | ① | 57.2±8.6 | e | 26 | 124 | 1,2,7,12,16 | 7 |
| Li et al. [22] | 2021 | China | 2015.06–2020.06 | ① | 56.7±6.58/57.16±6.62 | a | 60 | 60 | 7,12,22,27 | 8 |
| Liu et al. [23] | 2021 | China | 2013.06–2019.12 | ① | 58–79/58–78 | c | 62 | 223 | 3,7,12,28 | 7 |
| Liu et al. [24] | 2016 | China | 2013.06–2015.06 | ① | 40–75/38–74 | c | 38 | 334 | 1,2,3,6,12 | 7 |
| Liu et al. [25] | 2021 | China | 2020.07–12 | ① | 55.81±15.95 | f | 40 | 139 | 12,17,29,30 | 7 |
| Luo et al. [26] | 2019 | China | 2012.12–2017.12 | ① | —— | g | 49 | 379 | 1,2,6,7,12,13 | 7 |
| Lv et al. [27] | 2021 | China | 2014.06–2019.06 | ① | 58.62±8.73 | e | 32 | 66 | 2,6 | 7 |
| Ma et al. [28] | 2021 | China | 2016.10–2020.04 | ① | 60.14±10.56/49.82±8.75 | c | 41 | 219 | 2,6,11,31 | 7 |
| Ma et al. [29] | 2021 | China | 2018.07–2019.07 | ① | —— | h | 27 | 343 | 1,8,32,33 | 6 |
| Ran et al. [30] | 2018 | China | 2013.02–2017.03 | ① | 41.26±4.62/41.34±4.53 | c | 63 | 100 | 4,6,7,10,14,34 | 7 |
| Shen et al. [31] | 2020 | China | 2015.01–2019.12 | ① | —— | d | 24 | 1950 | 7 | 6 |
| Wan et al. [32] | 2014 | China | 2010.07–2013.09 | ① | 57.2±16.3 | b | 34 | 330 | 1,2,6,12 | 6 |
| Wang et al. [33] | 2020 | China | 2015.01–2020.05 | ① | 55.08±15.39 | h | 60 | 260 | 6,7,9,13,20,35,36 | 7 |
| Wang et al. [34] | 2019 | China | 2016.01–2017.12 | ① | 54.68±10.06 | h | 82 | 368 | 1,7,12,16 | 7 |
| Wang et al. [35] | 2022 | China | 2018.05–2021.05 | ① | —— | f | 16 | 74 | 2,6,7,9,12,19 | 7 |
| Wang et al. [36] | 2014 | China | 2011.01–2013.03 | ① | 42.72±10.85 | c | 37 | 203 | 1,2,7,8,12,16 | 7 |
| Wu et al. [37] | 2020 | China | 2018.01–2019.01 | ① | 61.39±4.95/61.04±5.02 | a | 100 | 100 | 6,7,37 | 8 |
| Xiao et al. [38] | 2018 | China | 2014.01–2015.01 | ① | 57.73±10.18/56.37±10.58 | b | 180 | 180 | 1,2,7,8,12,16 | 7 |
| Yuan et al. [39] | 2022 | China | 2019.12–2020.12 | ① | —— | g | 33 | 147 | 1,2,6,12,13 | 8 |

*(Continued)*

**Table 1.** (Continued)

| Study | Publish year | Country | Investigation time | Research type | Age Case group/ Control group | Diagnostic criteria for CRBSI | Number of cases | Number of controls | Rask factors | Scores |
|---|---|---|---|---|---|---|---|---|---|---|
| Zhang et al. [40] | 2018 | China | 2015.10–2017.10 | ① | 65.11±16.10 | e | 35 | 30 | 2,7,12 | 8 |
| Zhang et al. [41] | 2019 | China | 2014.01–2016.12 | ① | 59.92±16.10/ 57.39±14.02 | e | 38 | 350 | 2,8,12 | 7 |
| Zhang et al. [42] | 2016 | China | 2011.01–2013.01 | ① | 59.32±14.48/ 50.18±14.29 | b | 43 | 245 | 2,6,11.38,39,40 | 7 |
| Zhang et al. [43] | 2019 | China | 2016.01–2018.10 | ① | 63.34±14.00 | b | 58 | 40 | 2,12 | 7 |
| Zhao et al. [44] | 2021 | China | 2015.01–2019.01 | ① | 62.58±8.75 | e | 29 | 119 | 2,6,9,12,16 | 6 |
| Zhao et al. [45] | 2017 | China | 2012.01–2016.12 | ① | —— | e | 61 | 85 | 2,12,16,41,42,43 | 7 |
| Zheng et al. [46] | 2018 | China | 2014.11–2017.11 | ① | 50.43±12.56/ 54.43±10.36 | i | 72 | 72 | 2,20 | 9 |
| Zhou et al. [47] | 2020 | China | 2012.01–2019.01 | ① | 45.79±16.80 | e | 52 | 124 | 2,4,6,7,10,14, 44 | 7 |
| Fram et al. [48] | 2015 | Brazil | 2010.01–2013.06 | ① | 56±17.1/53±16.3 | j | 81 | 81 | 23,45 | 9 |
| Martin et al. [49] | 2020 | Australia | 2013.01–2018.06 | ① | 42–69/52–70 | k | 39 | 188 | 6,12 | 9 |
| R Çaylan et al. [50] | 2010 | Turkey | 2003.10–2006.10 | ① | 59.2±15.7/56.3 ±14.9 | e | 63 | 185 | 6,8,19,22,46,47 | 8 |
| Taylor et al. [51] | 2002 | Canada | 1998.12–1999.05 | ① | 17–89/22–91 | c | 93 | 93 | 17,21,48 | 9 |
| Grothe et al. [52] | 2010 | Brazil | ——— | ① | 14–94/17–78 | c | 94 | 62 | 6,7,12,18,49,50 | 9 |
| Hadian et al. [53] | 2020 | Iran | 2015.03–2018.03 | ② | 62.94±16.11/ 57.53±19.15 | c | 84 | 67 | 15 | 7 |
| Cheng et al. [54] | 2018 | China | 2010.04–2015.05 | ① | 41±17.60/41.2 ±16.7 | d | 57 | 114 | 4,6,7,10 | 8 |
| Samani et al. [55] | 2014 | Iran | 2012.03–2013.04 | ② | 53.5±15.4/48.6 ±13.2 | c | 44 | 176 | 7,8,21,51 | 9 |
| Donati et al. [56] | 2020 | Italy | 2011.01–2015.06 | ② | 75.3±12.3 | c | 16 | 63 | 1,15,52,53,54 | 9 |
| Lemaire et al. [57] | 2009 | France | 1982.11–2005.11 | ② | 16–95 | c | 226 | 1523 | 7,12,18,55 | 8 |
| Herc et al. [58] | 2017 | America | 2013.01–2016.10 | ② | ——— | e | 249 | 22839 | 17,56,57,58,59,60 | 8 |
| Martín-Peña et al. [59] | 2012 | Spain | 2014.09–2015.10 | ② | 20–89 | c | —— | —— | 22,61 | 9 |
| Zanoni et al. [60] | 2021 | Italy | 2014.02–2017.01 | ① | 56–78 | c | 84 | 329 | 6,22,62 | 9 |

(*Continued*)

**Table 1.** (Continued）

| Study | Publish year | Country | Investigation time | Research type | Age Case group/ Control group | Diagnostic criteria for CRBSI | Number of cases | Number of controls | Rask factors | Scores |
|---|---|---|---|---|---|---|---|---|---|---|
| Murea et al. [61] | 2014 | America | 2005.01– 2007.12 | ② | 59.9±15.7 | c | 200 | 264 | 1,6 | 7 |

Research type: ①Case-control study; ②Cohort study.

Diagnostic criteria for CRBSI: A. Guidelines for the Prevention and Treatment of Infections Associated with Intravascular Catheters; B. Diagnostic Criteria for Hospital Infections; C. Positive blood culture; D. Centers for Disease Control and Prevention 2002 Edition Diagnostic Criteria for Intravascular Catheter-Associated Bloodstream Infections; E. Diagnostic criteria for CRBSI developed by the Infectious Diseases Society of America in 2009/Centers for Disease Control and Prevention Clinical Practice Guidelines for the Diagnosis and Management of Intravascular Catheter-Associated Infections; F. 2019 Vascular Access Clinical Practice Guidelines; G. Expert Consensus on Vascular Access for Hemodialysis in China" issued by the Vascular Access Group of the Management Branch of Hemodialysis Center of China Hospital Association in 2014; H. 2017 U.S. Infection Prevention Guidelines; I. National Healthcare Safety Net Diagnostic Criteria developed by the Centers for Disease Control; J. according to the specific criteria of the National Healthcare Safety Network Dialysis Event Surveillance Manual (Centers for Disease Control and Prevention); K. KDOQI clinical practice guidelines and clinical practice recommendations for 2006 updates.

Risk factors: 1. Age; 2. Albumin(ALB); 3. Anemia; 4. Acute physiology and chronic health score(APACHE II scores); 5. C-reactive protein(CRP); 6. Catheter site; 7. Catheter duration; 8. Number of catheterizations; 9. Catheter type; 10. CD4$^+$ cell count; 11. Cholesterol; 12. Diabetes mellitus; 13. Dialysis duration; 14. External hospital tube; 15. Gender; 16. Hemoglobin(Hb); 17. History of catheter-associated infection; 18. Hypertension; 19. Inadequate hand hygiene; 20. Procalcitonin(PCT); 21. Poor patient hygiene; 22. Renal disease; 23. Venous access type; 24. Payment method; 25. White blood cell(WBC); 26. Catheter mode; 27. malignant tumor; 28. Smoke; 29. Number of nursing interventions; 30. Central venous disease; 31. Low-density lipoprotein cholesterol (LDL-C); 32. Renal replacement therapy time; 33. Nutritional status; 34. Mean arterial pressure(MAP); 35. NRL; 36. PTH; 37. Comorbidities of underlying diseases; 38. Low density lipoprotein, apolipoprotein AI; 39. Phosphorus (P); 40. Chlorine(Cl); 41. Neutrophil(NE); 42. Red blood cell(RBC); 43. Serum creatinine(Scr); 44. Central venous pressure(CVP); 45. Previous hospitalization; 46. Administration of antibiotics at the time of catheterization; 47. Emergency situation for catheter insertion; 48. Contiguous skin infection; 49. Hospitalization time; 50. Positive cultures from the catheter tip; 51. Poor hygiene of the medical staff; 52. Infections from other regions; 53. Peripheral obstructive arterial disease (POAD); 54. History of arteriovenous fistula (AVF); 55. Previous history of a bacteremic episode; 56. Receipt of TPN through the PICC; 57. Past or present history of hematological cancer; 58. Number of PICC lumens; 59. Active cancer with ongoing chemotherapy; 60. Presence of another CVC at the time of PICC placement; 61. Previous tunneled hemodialysis catheters; 62. Non-tunneled (NT) CVC.

that diabetes mellitus significantly increases the risk of CRBSI (OR = 2.52, 95% CI [1.95, 3.25], $P < 0.00001$). Fig 4 shows the forest plot for diabetes mellitus.

(4) Anemia and its effect on CRBSI:

Two studies in our analysis reported a relationship between anemia and the incidence of CRBSI in HD patients. A high level of statistical heterogeneity was noted ($I^2 = 92\%$, $P = 0.0005$), which prompted the use of a random-effects model. The meta-analysis indicated

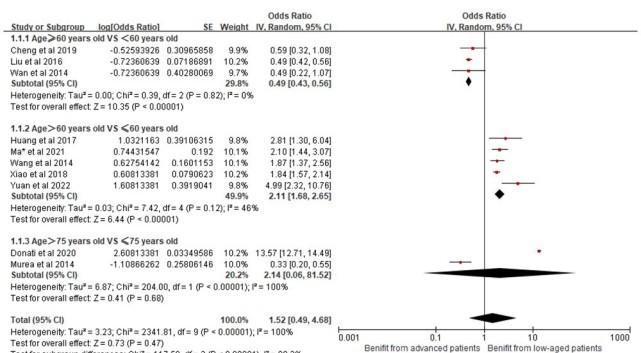

**Fig 2. Forest plot of the relationship between age and the occurrence of CRBSI in HD patients.**

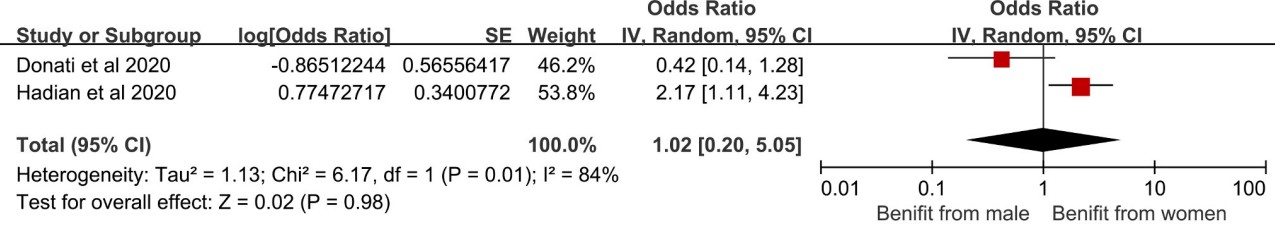

**Fig 3. Forest plot of the relationship between gender and the occurrence of CRBSI in HD patients.**

that anemia does not significantly impact the risk of CRBSI (OR = 1.36, 95% CI [0.15, 11.92], $P$ = 0.78). Fig 5 shows the forest plot for anemia.

(5) Renal disease and its effect on CRBSI:

Four studies in our dataset reported a correlation between renal disease and the incidence of CRBSI in HD patients. Due to the observed heterogeneity ($I^2$ = 66%, $P$ = 0.03), a random-effects model was employed for the meta-analysis. The findings showed that renal disease significantly escalates the risk of CRBSI (OR = 3.45, 95% CI [1.71, 6.96], $P$ = 0.0006). Fig 6 shows the forest plot for renal disease.

(6) History of catheter-associated infection and its effect on CRBSI:

Our study reviewed three articles that assessed the link between a history of catheter-associated infection and CRBSI in HD patients. Statistical heterogeneity across these studies was not

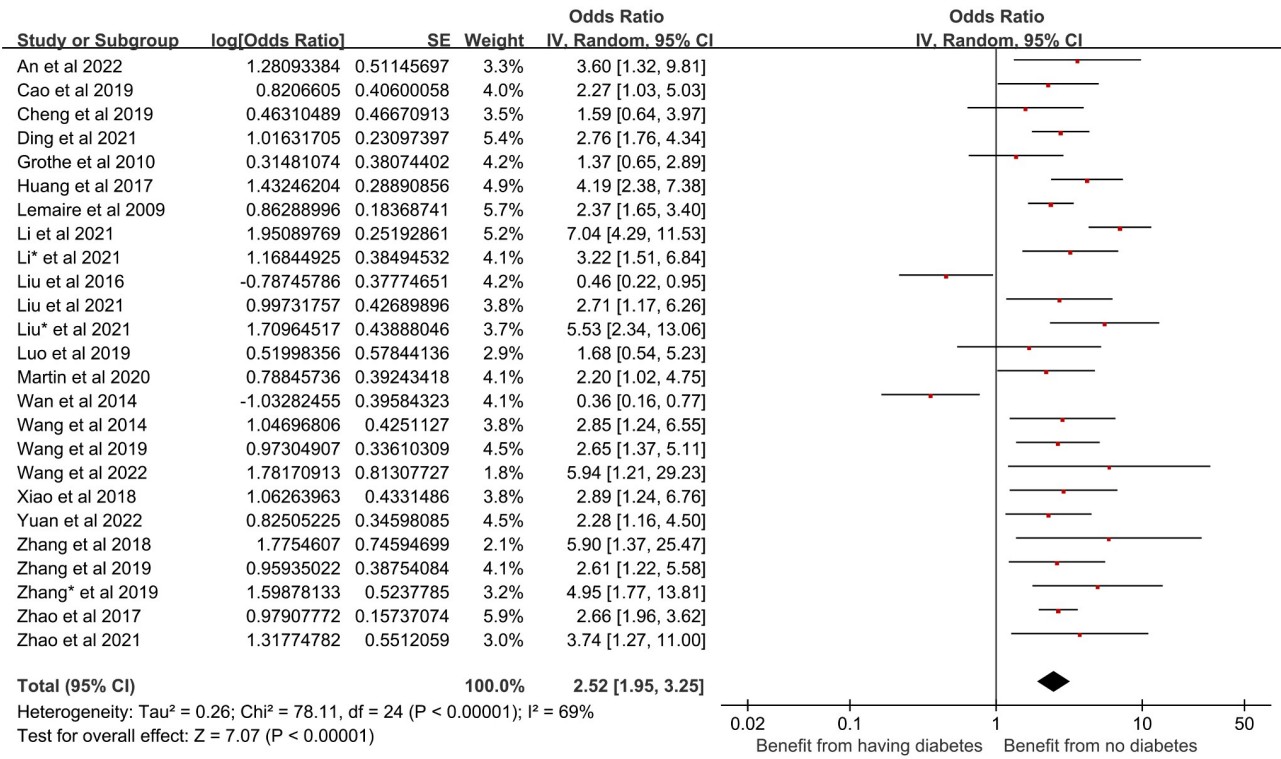

**Fig 4. Forest plot of the relationship between diabetes mellitus and the occurrence of CRBSI in HD patients.**

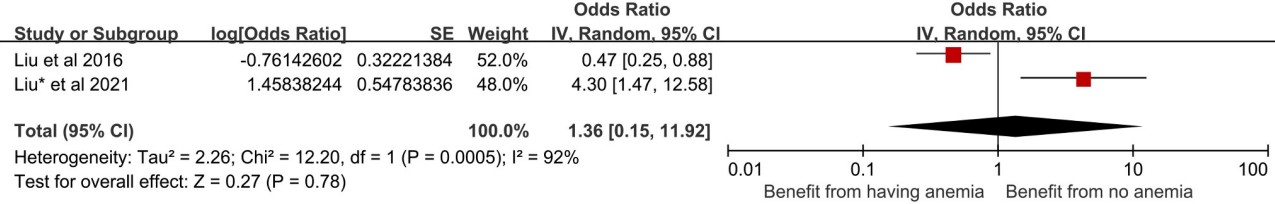

**Fig 5. Forest plot of the relationship between anemia and the occurrence of CRBSI in HD patients.**

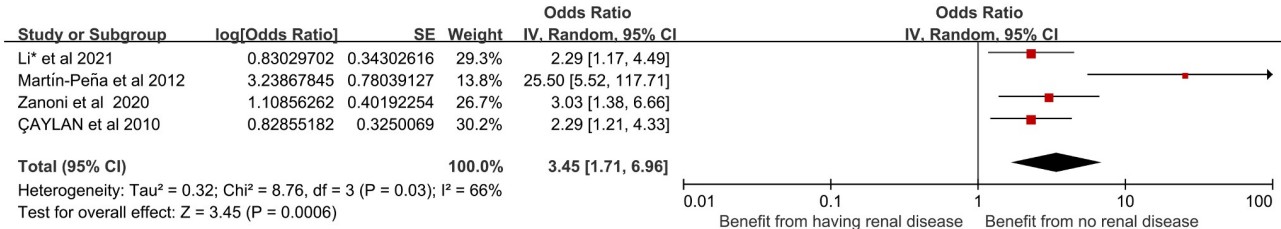

**Fig 6. Forest plot of the relationship between renal disease and the occurrence of CRBSI in HD patients.**

significant ($I^2$ = 30%, $P$ = 0.24), allowing for a fixed effects model in the meta-analysis. The analysis indicated that a history of catheter-associated infection significantly increases the risk of CRBSI (OR = 2.79, 95% CI [1.96, 3.98], $P < 0.00001$). Fig 7 shows the forest plot for history of catheter-associated infection.

(7) Hypertension and its effect on CRBSI:

Two studies within our review investigated the correlation between hypertension and CRBSI in HD patients. The lack of statistical heterogeneity among these studies ($I^2$ = 0%, $P$ = 0.59) justified the use of a fixed-effect model in our meta-analysis. The results identified hypertension as a significant risk factor for CRBSI (OR = 1.43, 95% CI [1.08, 1.91], $P$ = 0.01). Fig 8 shows the forest plot for hypertension.

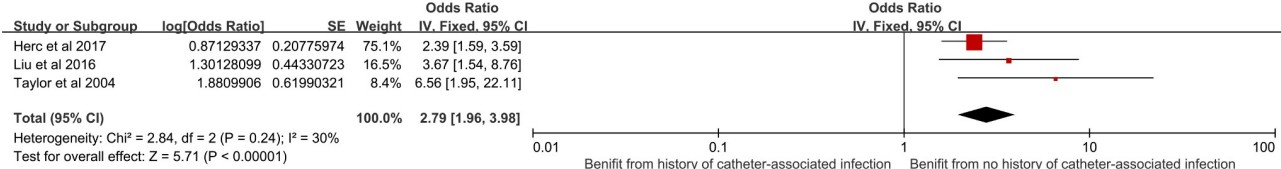

**Fig 7. Forest plot of the relationship between history of catheter-associated infection and the occurrence of CRBSI in HD patients.**

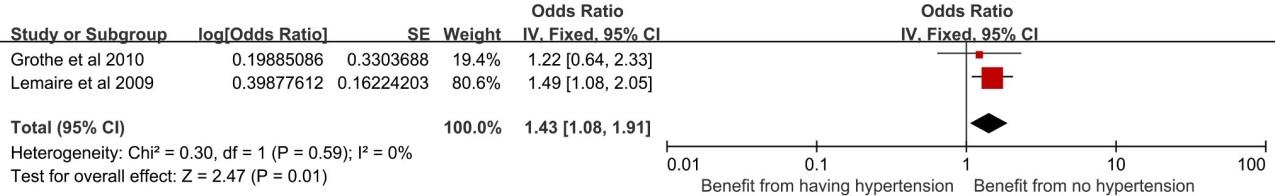

**Fig 8. Forest plot of the relationship between hypertension and the occurrence of CRBSI in HD patients.**

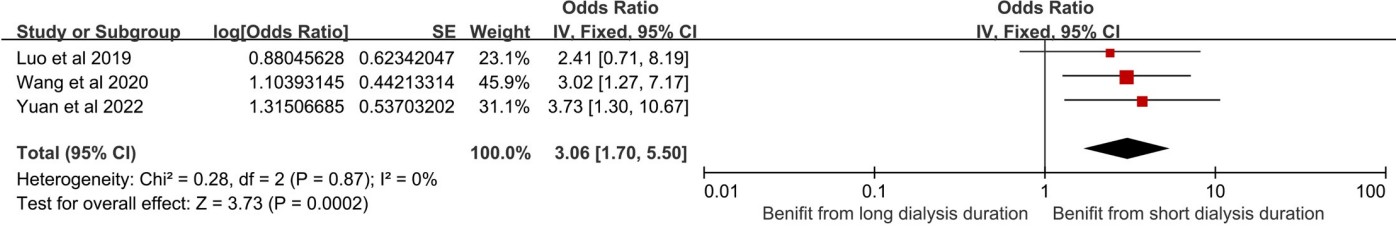

**Fig 9. Forest plot of the relationship between dialysis duration and the occurrence of CRBSI in HD patients.**

(8) Dialysis duration and its effect on CRBSI:

Three articles were analyzed to determine the relationship between dialysis duration and CRBSI in HD patients. The absence of statistical heterogeneity between these studies ($I^2 = 0\%$, $P = 0.87$) facilitated the use of a fixed effects model for meta-analysis. The results indicated a significant association, with longer dialysis duration increasing the risk of CRBSI (OR = 3.06, 95% CI [1.70, 5.50], $P = 0.0002$). Fig 9 shows the forest plot for dialysis duration.

(9) Catheter site and its effect on CRBSI:

In this review, 20 articles were examined for a correlation between catheter site and CRBSI in HD patients. However, significant statistical heterogeneity was observed among these studies ($I^2 = 81\%$, $P < 0.00001$), necessitating a random-effects model for meta-analysis. The analysis revealed that the catheter site significantly influences the risk of CRBSI (OR = 1.91, 95% CI [1.35, 2.70], $P = 0.0002$). Fig 10 shows the forest plot for catheter site.

(10) Catheter duration and its effect on CRBSI:

Twenty-two articles were studied to evaluate the association between catheter duration and CRBSI in HD patients. Subgroup analysis was performed for different catheter duration and

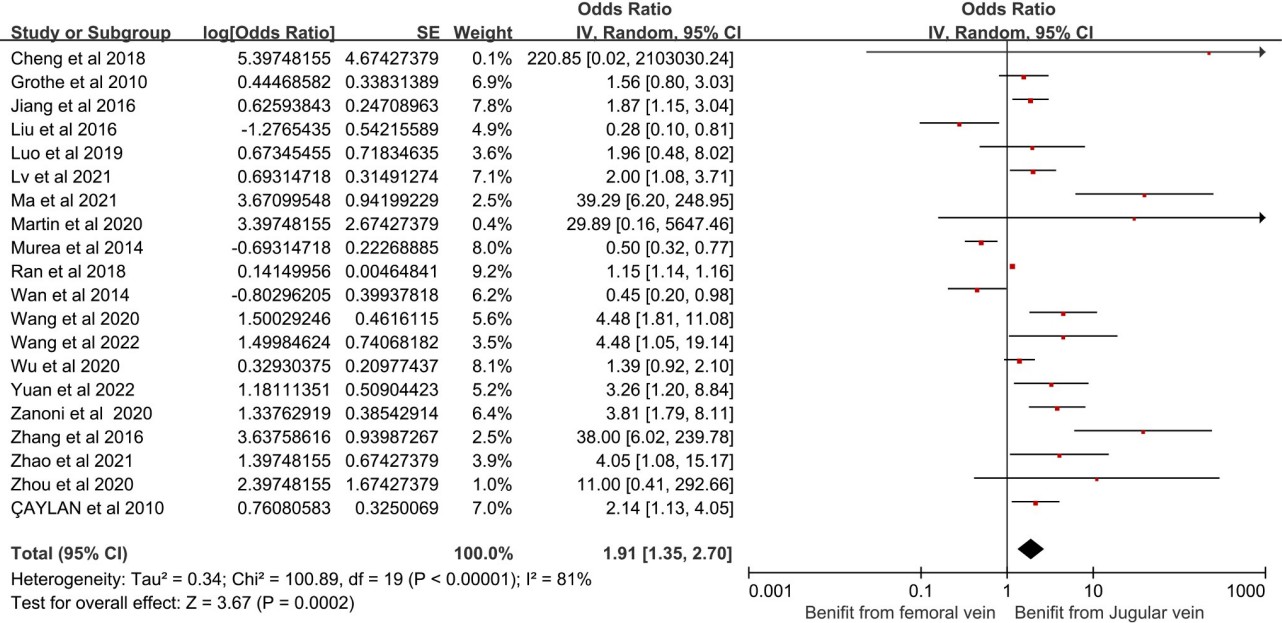

**Fig 10. Forest plot of the relationship between catheter site and the occurrence of CRBSI in HD patients.**

they were divided into six different groups: >7d and ≤7d, ≥14d and <14d, ≥15d and <15d, ≥30d and <30d, >90d and ≤90d, >1year and ≤1year. There was substantial statistical heterogeneity among these studies ($I^2$ = 97%, P < 0.00001), requiring a random-effects model for meta-analysis. The findings confirmed that the duration of catheter use significantly contributes to the risk of CRBSI(OR = 2.06, 95% CI [1.17, 3.60], P < 0.0001). Fig 11 shows the forest plot for catheter duration.

(11) Number of catheterizations and their effect on CRBSI:

Eight studies included in our analysis reported a significant relationship between the number of catheterizations and the incidence of CRBSI in HD patients. Subgroup analysis was performed for different number of catheterizations and they were divided into two different groups: ≥2 times and <2 times, ≥3 times and <3 times. The analysis revealed that the number

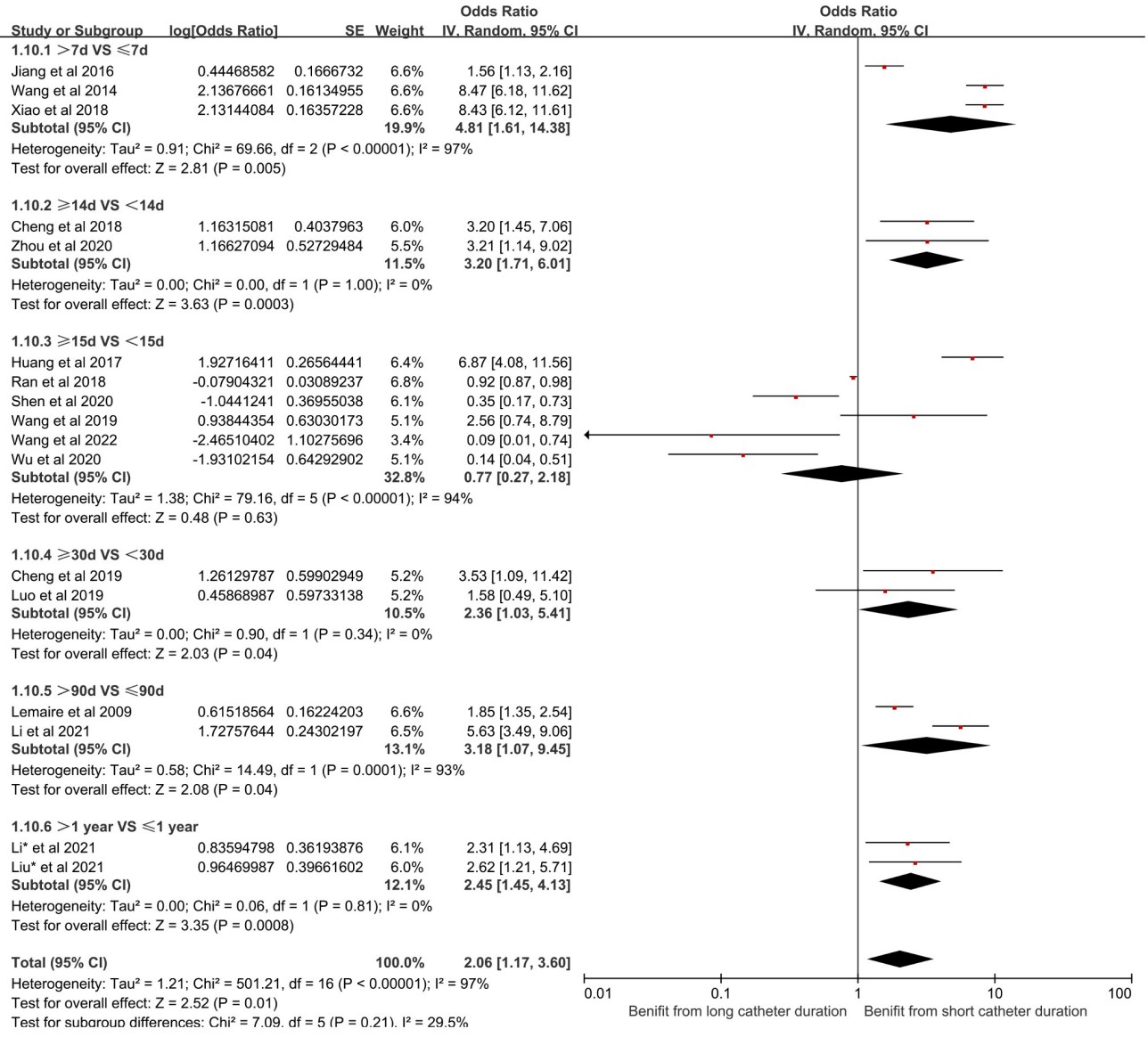

**Fig 11. Forest plot of the relationship between catheter duration and the occurrence of CRBSI in HD patients.**

of catheterizations significantly influences the risk of CRBSI, and different number of catheterizations is the main source of heterogeneity. Fig 12 shows the forest plot for the number of catheterizations.

(12) Catheter type and its effect on CRBSI:

Four studies investigated the association between catheter type and CRBSI among HD patients, presenting a low level of statistical heterogeneity ($I^2 = 16\%$, $P = 0.31$). Accordingly, a fixed effects model was utilized for meta-analysis. The results indicated that catheter type is a significant risk factor for CRBSI (OR = 3.83, 95% CI [2.13, 6.87], $P < 0.00001$). Fig 13 shows the forest plot for catheter type.

(13) External hospital tube and its effect on CRBSI:

Two studies reported a relationship between the use of external hospital tube and CRBSI in HD patients. Notable statistical heterogeneity was detected across these studies ($I^2 = 79\%$, $P = 0.03$). Our meta-analysis, conducted using a random-effects model, indicated that the use of external hospital tubes was not significantly associated with an increased risk of CRBSI (OR = 1.09, 95% CI [0.70, 1.70], $P = 0.70$). Fig 14 shows the forest plot for external hospital tube.

(14) CD4+ cells and their effect on CRBSI:

Three studies explored the association between CD4+ cells and CRBSI among HD patients, showing statistical heterogeneity ($I^2 = 78\%$, P = 0.010). Meta-analysis results showed that

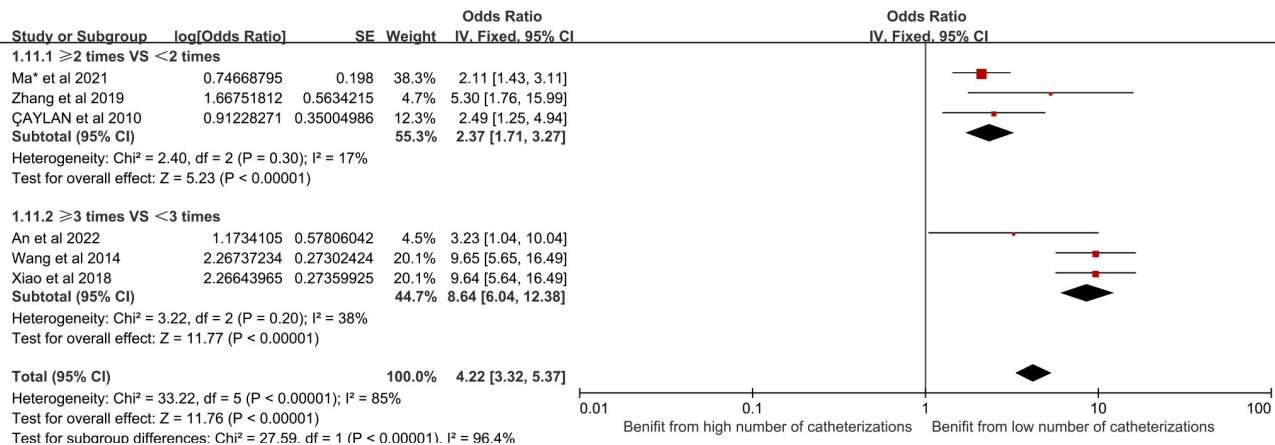

**Fig 12. Forest plot of the relationship between the number of catheterizations and the occurrence of CRBSI in HD patients.**

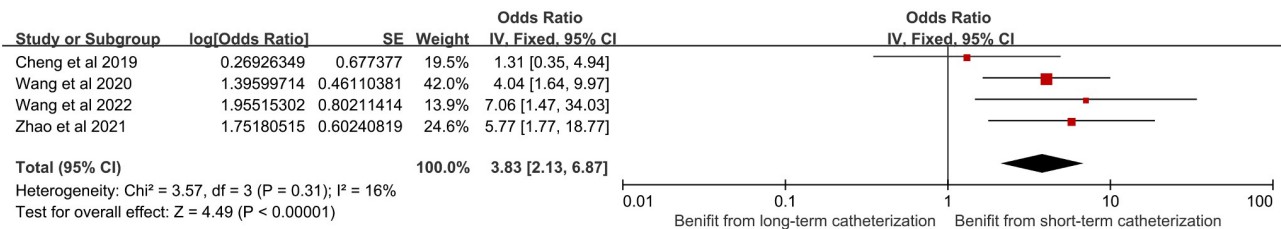

**Fig 13. Forest plot of the relationship between catheter type and the occurrence of CRBSI in HD patients.**

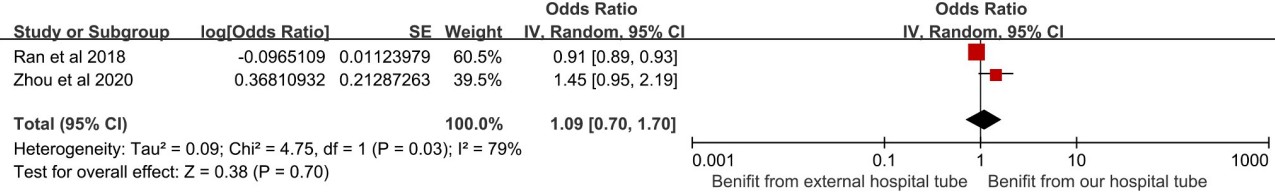

**Fig 14. Forest plot of the relationship between external hospital tube and the occurrence of CRBSI in HD patients.**

(OR = 0.52, 95% CI [0.23, 1.16], P = 0.11). After sensitivity analysis, the studies by Ran et al. 2018 was the main source of heterogeneity[30], and after excluding this study, we found that a lower CD4[+] cell count significantly increased the risk of CRBSI (OR = 0.33, 95% CI [0.18, 0.63], P = 0.0007). Fig 15 shows the forest plot for CD4+ cells, and Fig 16 shows sensitivity analysis on CD4[+] cells.

(15) Albumin (ALB) and its effect on CRBSI:

In this analysis, 22 studies reported a significant relationship between serum ALB and CRBSI in HD patients. Subgroup analysis was performed for different serum ALB levels and they were divided into three different groups: <30g/L and ≥30g/L, <35g/L and ≥35g/L, <40g/L and ≥40g/L. We discovered that low ALB levels significantly increased the risk of CRBSI. Fig 17 shows the forest plot for serum ALB.

(16) C-reactive protein (CRP) and its effect on CRBSI:

Two studies included in our analysis reported an association between CRP and the incidence of CRBSI in HD patients, with no significant statistical heterogeneity observed ($I^2 = 0\%$, $P = 0.98$). We applied a fixed-effect model for meta-analysis. Our results demonstrated a significant association, suggesting that elevated CRP levels serve as a risk factor for CRBSI (OR = 1.73, 95% CI [1.47, 2.03], $P < 0.00001$). Fig 18 shows the forest plot for CRP.

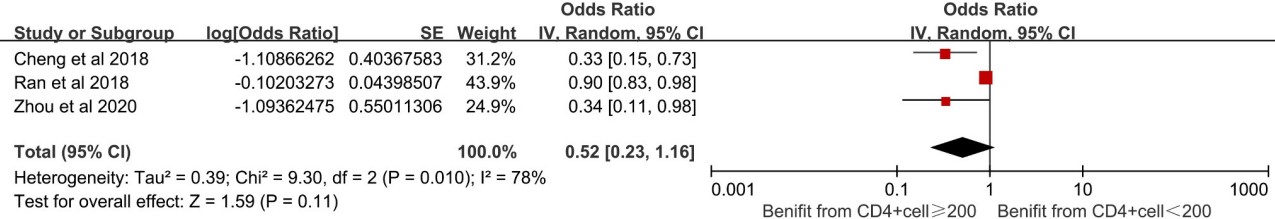

**Fig 15. Forest plot of the relationship between CD4+ cells and the occurrence of CRBSI in HD patients.**

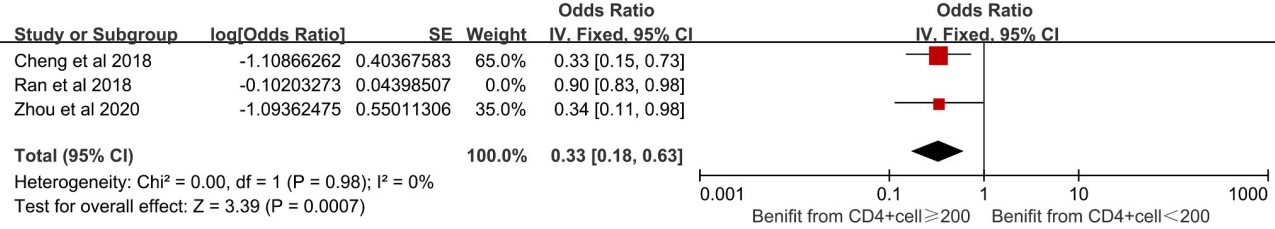

**Fig 16. Sensitivity analysis on CD4+ cells.**

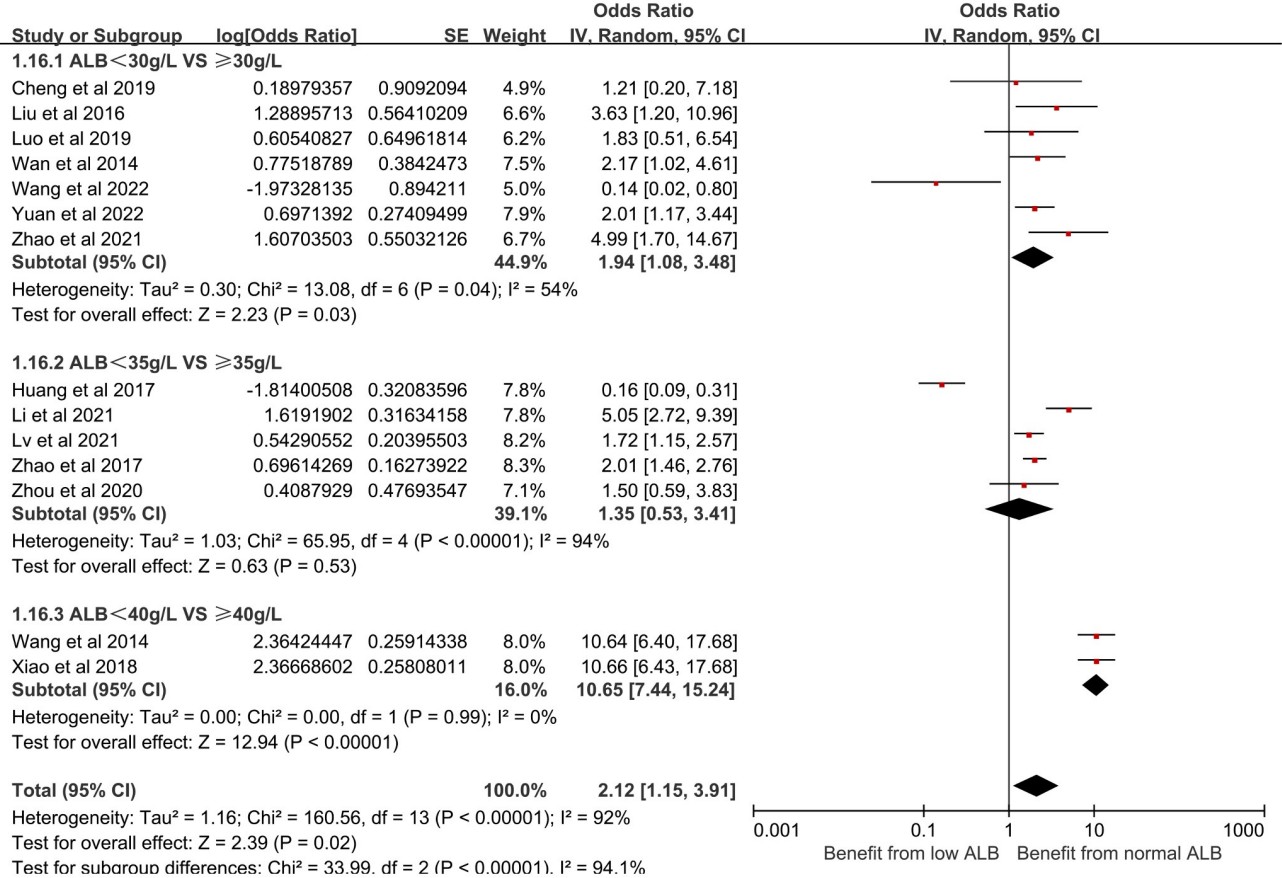

**Fig 17. Forest plot of the relationship between serum ALB and the occurrence of CRBSI in HD patients.**

(17) Hemoglobin (Hb) and its effect on CRBSI:

Six studies incorporated in our analysis reported a relationship between Hb and the development of CRBSI in HD patients. Subgroup analysis was performed for different Hb levels and they were divided into two different groups: <90g/L and ≥90g/L, <100g/L and ≥100g/L. These studies demonstrated substantial statistical heterogeneity ($I^2$ = 94%, P < 0.00001),using a random-effects model for meta-analysis. Our findings indicated that Hb levels are a significant risk factor for CRBSI (OR = 1.48, 95% CI [0.54, 4.07], $P$ = 0.45). Fig 19 shows the forest plot for Hb.

(18) Cholesterol and its effect on CRBSI:

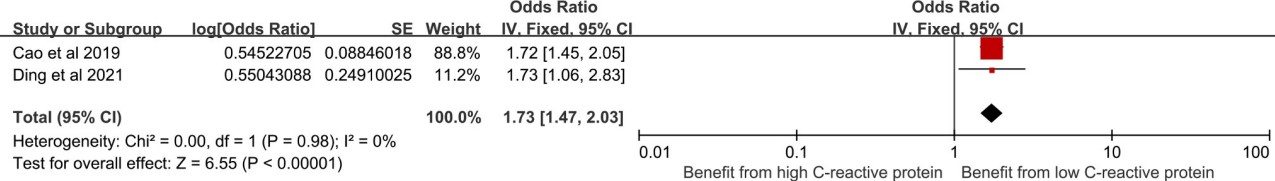

**Fig 18. Forest plot of the relationship between CRP and the occurrence of CRBSI in HD patients.**

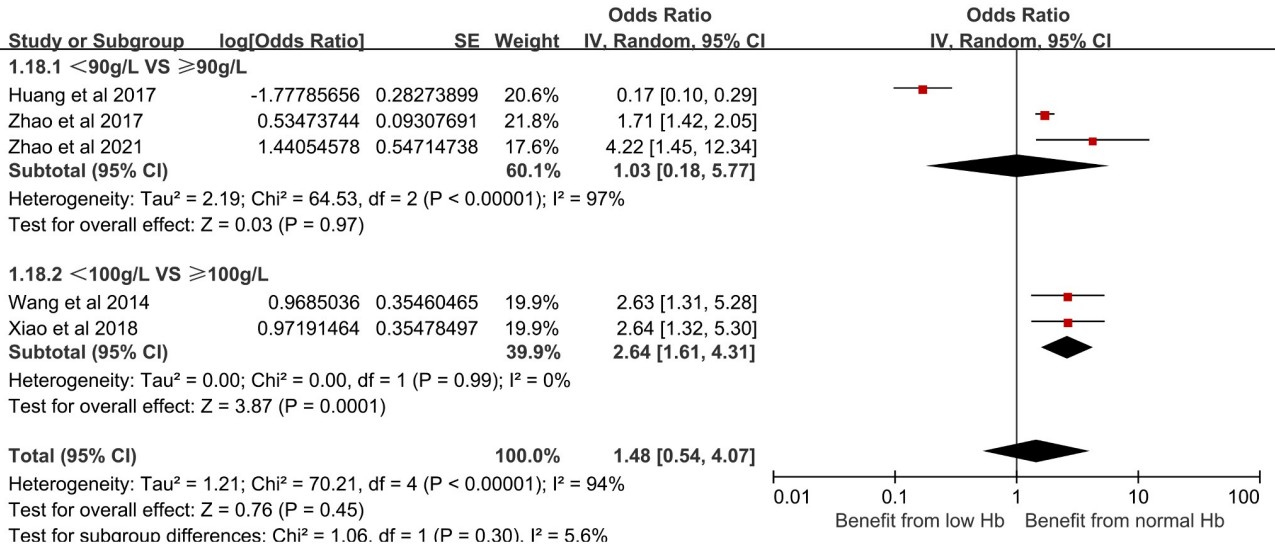

**Fig 19. Forest plot of the relationship between Hb and the occurrence of CRBSI in HD patients.**

Two studies in our review associated cholesterol levels with CRBSIs in HD patients, revealing considerable statistical heterogeneity ($I^2$ = 97%, $P$ < 0.00001). We consequently employed a random-effects model for the meta-analysis. Our results indicated that cholesterol levels were not a significant risk factor for CRBSIs (OR = 1.00, 95% CI [0.01, 91.05], $P$ = 1.00). Fig 20 shows the forest plot for cholesterol.

(19) Procalcitonin (PCT) and its effect on CRBSI:

Two studies in our analysis reported an association between PCT and the incidence of CRBSIs in HD patients, with no significant statistical heterogeneity identified ($I^2$ = 0%, $P$ = 0.44). We used a fixed-effect model for meta-analysis. Our findings demonstrated a significant association, suggesting PCT as a risk factor for CRBSIs (OR = 1.05, 95% CI [1.03, 1.06], $P$ < 0.00001). Fig 21 shows the forest plot for PCT.

(20) Inadequate hand hygiene and its effect on CRBSI:

Two studies in our review associated inadequate hand hygiene practices with the development of CRBSIs in HD patients, showing statistical heterogeneity ($I^2$ = 57%, $P$ = 0.13). We therefore applied a random-effects model for meta-analysis. Our analysis demonstrated that inadequate hand hygiene is a significant risk factor for CRBSIs (OR = 5.32, 95% CI [1.07, 26.37], $P$ = 0.04). Fig 22 shows the forest plot for inadequate hand hygiene.

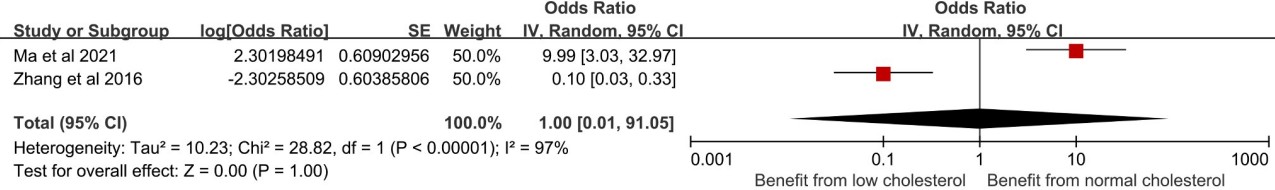

**Fig 20. Forest plot of the relationship between cholesterol and the occurrence of CRBSI in HD patients.**

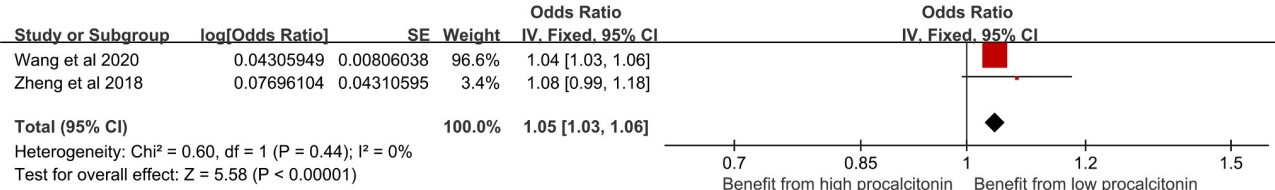

**Fig 21. Forest plot of the relationship between PCT and the occurrence of CRBSI in HD patients.**

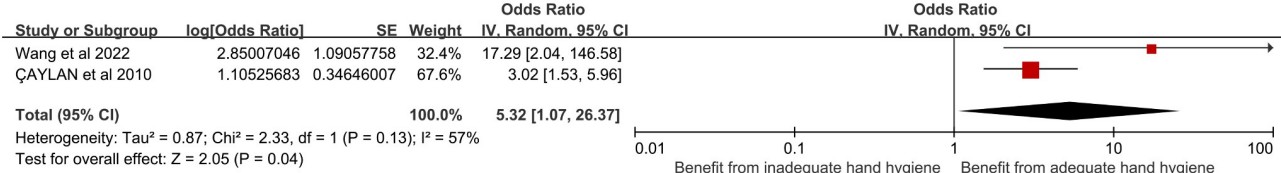

**Fig 22. Forest plot of the relationship between inadequate hand hygiene and the occurrence of CRBSI in HD patients.**

(21) and their APACHE II scores effect on CRBSI:

Three studies in our review reported an association between APACHE II scores and the incidence of CRBSIs in HD patients, showing statistical heterogeneity ($I^2$ = 82%, $P$ = 0.004). Meta-analysis results showed that (OR = 1.58, 95% CI [0.70, 3.59], P = 0.27). After sensitivity analysis, the studies by Ran et al. 2018 was the main source of heterogeneity [30], and after excluding this study, it can be concluded that elevated APACHE II scores are a significant risk factor for CRBSIs (OR = 2.41, 95% CI [1.33, 4.37], P = 0.004). Fig 23 shows the forest plot for APACHE II scores, and Fig 24 shows sensitivity analysis on APACHE II scores.

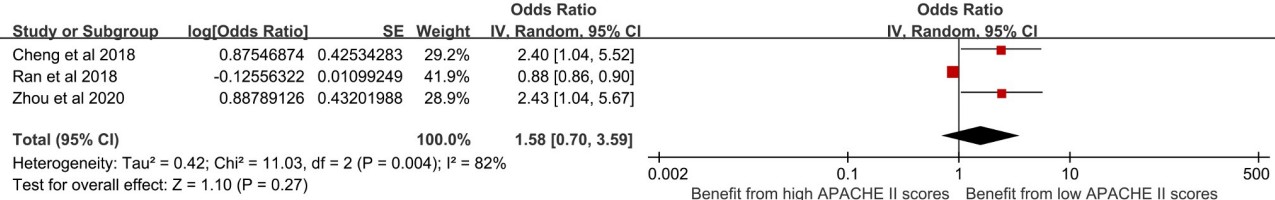

**Fig 23. Forest plot of the relationship between APACHE II scores and the occurrence of CRBSI in HD patients.**

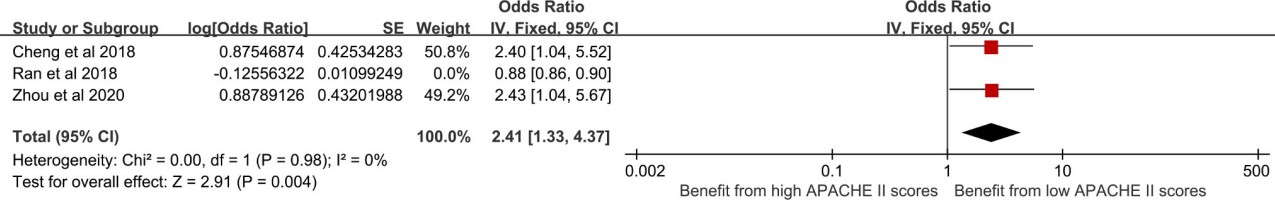

**Fig 24. Sensitivity analysis on APACHE II scores.**

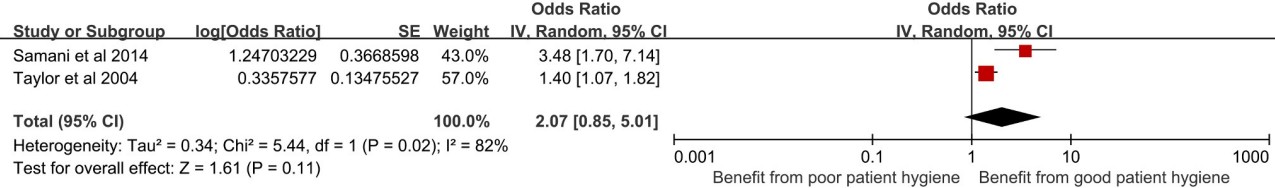

**Fig 25. Forest plot of the relationship between poor patient hygiene and the occurrence of CRBSI in HD patients.**

(22) Poor patient hygiene and its effect on CRBSI:

Two studies included in our analysis reported a relationship between poor patient hygiene and the development of CRBSIs in HD patients, indicating significant statistical heterogeneity ($I^2$ = 82%, $P$ = 0.02). We therefore employed a random-effects model for meta-analysis. Our results suggested that poor patient hygiene might act as a risk factor for CRBSIs, although the association was not statistically significant (OR = 2.07, 95% CI [0.85, 5.01], $P$ = 0.11). Fig 25 shows the forest plot for poor patient hygiene.

## 3.4 Sensitivity analysis results

A sensitivity analysis was conducted, employing both fixed-effect and random-effect models to evaluate and juxtapose the results associated with the research factors incorporated. These findings are presented in Table 2.

**Table 2. Odds ratio values and 95% CI for fixed and random-effects model.**

| Factor | Fixed-effect model | Random-effect model |
|---|---|---|
| Age | 5.52 (5.24,5.83) | 1.52(0.49,4.68) |
| Gender | 1.40 (0.79,2.49) | 1.02(0.20,5.05) |
| Diabetes mellitus | 2.59 (2.27,2.95) | 2.52 (1.95,3.25) |
| Anemia | 0.83(0.48,1.42) | 1.36 (0.15,11.92) |
| Renal disease | 2.86 (1.94,4.20) | 3.45 (1.71,6.96) |
| History of catheter-associated infection | 2.79 (1.96,3.98) | 3.09 (1.86,5.13) |
| Hypertension | 1.43 (1.08,1.91) | 1.43 (1.08,1.91) |
| Dialysis duration | 3.06 (1.70,5.50) | 3.06 (1.70,5.50) |
| Catheter site | 1.15 (1.14,1.16) | 1.91 (1.35,2.70) |
| Catheter duration | 1.16 (1.10,1.23) | 2.06 (1.17,3.60) |
| Number of catheterizations | 4.22 (3.32,5.37) | 4.55 (2.32,8.90) |
| Catheter type | 3.83 (2.13,6.87) | 3.81 (1.99,7.30) |
| External hospital tube | 0.91 (0.89,0.93) | 1.09 (0.70,1.70) |
| CD4$^+$ cell | 0.33 (0.18,0.63) | 0.33 (0.18,0.63) |
| ALB | 2.48(2.11,2.91) | 2.12 (1.15,3.91) |
| C-reactive protein | 1.73 (1.47,2.03) | 1.73 (1.47,2.03) |
| Hb | 1.50 (1.28,1.76) | 1.48 (0.54,4.07) |
| Cholesterol | 0.98 (0.42,2.27) | 1.00 (0.01,91.05) |
| PCT | 1.05 (1.03,1.06) | 1.05 (1.03,1.06) |
| Inadequate hand hygiene | 3.54 (1.86,6.77) | 5.32 (1.07,26.37) |
| APACHE II scores | 2.41 (1.33,4.37) | 2.41 (1.33,4.37) |
| Poor patient hygiene | 1.56 (1.22,2.00) | 2.07 (0.85,5.01) |

## 4 Discussion

### 4.1 Evaluation of methodological quality in included studies

Our analysis encompassed 49 studies, composed of 42 case-control studies and 7 cohort studies. All the studies explicitly stipulated the inclusion and exclusion criteria, along with the disease diagnostic parameters. With the quality of these studies being at the medium level or above, we assert that the results of this meta-analysis bear a considerable level of reliability.

### 4.2 Relationship between risk factors and CRBSI

Our meta-analysis of all the elements culminated in identifying 17 risk factors associated with the incidence of CRBSI in HD patients. These include age, diabetes mellitus, renal disease, history of catheter-associated infection, hypertension, dialysis duration, catheter site, catheter duration, number of catheterizations, catheter type, $CD4^+$ cell, ALB, CRP, Hb, PCT, inadequate hand hygiene, and APACHE II scores. The effects of these risk factors on CRBSI are elucidated below:

**4.2.1 General patient characteristics.** *Age and CRBSI*. Research indicates that the probability of developing CRBSI escalates with age. This increase could be attributed to the concomitant decline in organ functionality and immunological resistance that accompanies aging, thereby augmenting infection susceptibility [62]. Additionally, vascular elasticity diminishes with age, increasing the brittleness and susceptibility to damage, thus heightening the risk of infection [63]. A study have reported that elderly HD patients exhibit a CRBSI risk 2.22 times higher than their younger counterparts [64]. In addition, elderly patients often have multiple chronic diseases, therefore, special attention should be paid to the prevention and management of CRBSI in elderly HD patients, to improve patient prognosis, and enhance their quality of life.

**4.2.2 Disease history.**

(1) The impact of diabetes mellitus on CRBSI

Patients with persistently elevated blood glucose levels tend to experience severe impairment across various bodily systems. These patients are at a heightened risk of developing CRBSI. The pathogenesis might be attributable to tissue damage due to abnormal glucose metabolism, resulting in microvascular lesions. Glucose-rich tissues provide an environment conducive to the proliferation of pathogenic bacteria, thereby leading to infection [65]. Alternatively, the reduced cytokine secretion in patients with chronic hyperglycemia impairs cellular immune function and diminishes the body's resistance, facilitating bacterial invasion and increasing infection risk. Hence, it is crucial to enhance blood glucose management in diabetic patients in clinical practice, aiming to maintain stable blood glucose levels, bolster resistance, and thereby reduce the incidence of CRBSI.

(2) The impact of renal disease on CRBSI:

Patients with end-stage renal disease present an elevated risk of developing CRBSI. This increase can be attributed to multiple factors, including advanced disease severity, concurrent complications, and compromised immune status [22]. Furthermore, renal disease patients typically experience a reduction in their immune system functionality, leading to decreased urine output. This reduced urinary flushing effect may facilitate the persistence of urinary tract bacteria, thereby heightening the infection risk.

(3) History of catheter-associated infections and their impact on CRBSI:

In patients with prior incidences of catheter-associated infection, diminished bodily resistance and immune function can escalate the susceptibility to bacterial invasion, thereby increasing the likelihood of subsequent infections. Owing to a high rate of infection and low rate of resolution, treatment of these patients is both challenging and complex. Consequently, it is crucial to enforce stringent infection control measures for patients with a history of catheter-associated infection to ensure their safety.

(4) The impact of hypertension on CRBSI:

The primary role of the kidney is to facilitate urine production and the elimination of metabolic waste products from the body. However, patients with chronic kidney disease often exhibit delayed removal of excess water due to declining kidney function. This delay results in water and sodium retention, triggering the onset of hypertension. For patients undergoing HD, elevated blood pressure significantly increases the risk of fatal cardiovascular events—up to eight times more than in the general population—thereby adversely impacting patient prognosis and quality of life [66]. Additionally, chronic hypertension can contribute to the development of CRBSI. It is likely due to long-term hypertension-induced vascular wall damage and reduced vascular elasticity, which can cause vessel rupture during catheter placement, subsequently leading to infection.

### 4.2.3 Evaluation of laboratory indicators.

(1) The impact of albumin on CRBSI:

Serum albumin deficiency is an identified risk factor for malnutrition in patients undergoing HD. This condition, termed hypoproteinemia, is commonly seen in HD patients with chronic kidney failure. The primary causes are threefold: (1) During the HD process, the loss of proteins and amino acids can accelerate protein decomposition, leading to a negative nitrogen balance and resultant malnutrition, especially when accompanied by inadequate energy intake. (2) A compromised digestive system can hinder protein and energy absorption, exacerbating malnutrition. (3) Long-term chronic kidney failure patients often adopt a low-protein diet to mitigate disease progression, further predisposing them to malnutrition over the course of their extended treatment duration. Consequently, it is critical to intensify nutritional monitoring in long-term HD patients, with an emphasis on high-quality protein consumption. In cases of severe hypoproteinemia or eating difficulties, albumin infusion may be necessary to bolster patient resistance to prevent the incidence of CRBSI.

(2) The impact of hemoglobin on CRBSI:

Hemoglobin, similar to serum albumin, is a crucial indicator of nutritional status. Suboptimal hemoglobin levels can impair immunity, thereby predisposing patients to infections. This susceptibility to infection arises because a diminished red blood cell count reduces the body's capacity to combat pathogenic bacteria, thereby increasing the likelihood of infection.

(3) The impact of procalcitonin on CRBSI:

PCT can serve as a useful biomarker for systemic infection. It is instrumental not only in the early diagnosis of bacterial infections but also in assessing infection severity and differentiating among various types of infections [67]. Patients with elevated PCT levels are at higher risk of infection. A study by Yasemin demonstrated that patients who met the diagnostic criteria for CRBSI had significantly elevated PCT levels in the infection group compared to the non-infected group [68]. Numerous international studies corroborate the value of PCT in early CRBSI diagnosis [69, 70]. In clinical practice, early CRBSI diagnosis can be facilitated by an integrated approach that combines patient clinical symptoms and signs with PCT levels,

thus enabling timely initiation of anti-infective treatments, early intervention, and prevention of disease progression.

(4) The impact of CD4$^+$ cell count on CRBSI:

Research indicates that a CD4$^+$ cell count of less than 200 cells/μl is suggestive of significant immunosuppression in patients. This immunosuppression may be due to repeated usage and long-term indwelling of the catheter, which facilitates bacterial invasion and colonization of the catheter's inner surface. This, in turn, increases the likelihood of bacterial entry into the body, leading to sustained physiological stress, diminished immune response, and consequently a higher infection incidence. As such, it is crucial to bolster clinical prevention strategies and control measures for these patients to mitigate the risk of CRBSI [1, 47].

(5) The impact of C-reactive protein on CRBSI:

The assessment of CRP levels has been found to contribute substantially to early identification of catheter-associated infections in hemodialysis patients, showing superior sensitivity and specificity compared to conventional infection markers [71]. Studies indicate an increased risk of CRBSI in hemodialysis patients corresponding to elevated CRP levels. This could be attributed to the fact that high CRP levels may reduce catheter blood flow, thereby promoting thrombus formation and subsequently increasing the infection probability [14].

**4.2.4 Catheter-related factors.**

(1) The impact of catheter site on CRBSI:

Blood purification is a critical treatment approach for contemporary kidney disease patients, making the establishment of effective vascular access a priority. The most desirable vascular access for hemodialysis patients is an autologous arteriovenous fistula. However, in cases where the fistula is immature or the patient's vascular condition is poor, central venous catheter insertion becomes necessary, which carries a higher risk of infection and consequently an elevated patient mortality rate [72]. Available research suggests that, compared to hemodialysis patients with arteriovenous fistulae, those with central venous catheterization face a 15-fold increase in infection risk [73]. For a majority of patients suffering from chronic renal failure due to various reasons, a temporary central line is typically established at the commencement of dialysis. A single-center study demonstrated that temporary central venous catheters constituted 81.26% of vascular access used in first-time hemodialysis [74], while another study reported the use of temporary central venous catheters in 76.8% of patients undergoing initial dialysis [75].

For patients with femoral vein catheterization, the risk of catheter-associated infections appears to be higher. This may be attributable to the anatomical location of the femoral vein, situated close to the perineum, a region typically characterized by a more humid environment conducive to bacterial proliferation. Further, the relatively slower blood flow within the femoral vein may promote thrombosis. The resultant conditions favor bacterial growth, thereby potentially contributing to infection. Consequently, it is crucial for patients with femoral vein catheters to maintain dry perineal skin [19]. Prior research has demonstrated that the risk of CRBSI is nearly 11 times higher with femoral as compared to jugular vein catheterization [76]. Chen Juan et al. further found a higher incidence of CRBSI among patients with femoral catheterization (35.1%) in comparison to those with jugular catheterization [77].

(2) The impact of catheter duration on CRBSI:

There is a positive correlation between catheter dwell time and the incidence of CRBSI. Essentially, the risk of infection escalates with an extended duration of catheterization. This

can be attributed to two primary factors. Firstly, during the course of prolonged indwelling catheter, the substantial impact of blood flow tends to damage the vascular endothelium, thereby inciting venous inflammation [78]. Secondly, the extended duration of catheterization exposes patients to repeated episodes of pollutant exposure. With time, the bacteria residing around the catheter undergo continuous growth and multiplication. When bacterial growth surpasses the body's infection threshold, a substantial release of bacteria into the bloodstream ensues, leading to bacteremia and the subsequent manifestation of clinical symptoms of infection. Medical staff should strengthen their knowledge of catheter maintenance, evaluate the necessity of catheter retention daily and remove them as soon as possible when not needed.

(3) The impact of catheter type on CRBSI:

The association between catheter type and the pathogenesis of CRBSI is complex. After 24 hours post-catheterization, a fibrin sheath encapsulates the catheter. This scenario paves the way for the exponential growth and colonization of microorganisms, thereby elevating the bacterial colonization rate. Concurrently, thrombus formation within the catheter permits bacterial colonies from the bloodstream to establish residence along the catheter wall. This dual mechanism is primarily responsible for the onset of infection.

(4) The impact of the number of catheterizations on CRBSI:

Several factors contribute to the high frequency of catheterization among patients. The main culprits include delayed patient referral, absence of durable dialysis access, suboptimal catheter functionality, patient's lack of knowledge regarding catheter maintenance, and recurrent instances of catheter stenosis, occlusion, or thrombosis. With each additional catheter insertion, the potential for vascular damage escalates due to the cumulative trauma inflicted by repeated punctures over an extended period. This leads to a subsequent increase in the infection incidence.

**4.2.5 Dialysis-related factors.** *Impact of dialysis duration on CRBSI.* There is a direct correlation between the duration of dialysis treatment and the likelihood of acquiring a CRBSI. As patients undergo more extended HD procedures, the probability of issues such as recurrent vascular access and dialysis machine sanitation escalates, thereby augmenting the risk of infection [79, 80].

**4.2.6 Factors pertaining to clinical practices.**

(1) The impact of inadequate hand hygiene on CRBSI:

Numerous studies have established the lack of adequate hand hygiene as a significant risk factor for the occurrence of CRBSI in HD patients. This is attributed to the invasive nature of central venous catheterization. If the healthcare professional does not strictly adhere to hand hygiene protocols during the procedure, it may facilitate the introduction of pathogenic bacteria into the bloodstream via the catheter, thereby enhancing the susceptibility to catheter infection [81].

(2) The impact of APACHE II scores on CRBSI:

The APACHE II scoring system, which includes the acute physiological score within 24 hours of admission, age, and chronic health evaluation, is predominantly employed for assessing severe clinical conditions and prognostication. A higher APACHE II score denotes a more critical health status of the patient. The incidence of CRBSI has been found to be intimately linked to the severity of the underlying disease. Consequently, as the APACHE II score rises, so does the incidence of CRBSI. This underscores the necessity for heightened preventive measures in critically ill and immunocompromised patients to curtail the prevalence of CRBSI [47].

**4.2.7 The significance of risk factors.** The risk factors selected may be influenced by the research purpose and background. Chinese studies usually focus more on laboratory indicators such as Albumin, Cholesterol, Hemoglobin and Albumin Cholesterol Hemoglobin. Because these indicators are more common and easy to measure in the clinic, they can provide more objective data. International researchers pay more attention to the overall situation and treatment effect of patients. In addition to studying common laboratory indicators and patient-related factors, they also conduct in-depth analysis from three aspects: catheter factors, infection factors and specific past medical history. Therefore, we can have a more comprehensive understanding of the risk factors of CRBSI in HD patients, which provides more theoretical guidance and basis for doctors and nurses to prevent CRBSI in HD patients. This allows us to target interventions to reduce the incidence of infection. And it is of great significance for improving the therapeutic effect of patients.

## 4.3 Significance to clinical practice

The influencing factors and related data of CRBSI obtained in this study have many application values: First, these findings can provide a reliable basis for medical personnel to formulate personalized prevention strategies. Secondly, these data can be used to develop risk assessment tools and monitoring indicators for CRBSI, identify high-risk patients through risk stratification and adjust prevention strategies accordingly to improve prevention effectiveness. In addition, these findings could be used to develop policy guidelines to regulate clinical practice. Minimize the incidence of CRBSI by providing standardized procedures and preventive measures. These policy guidelines can provide reference for medical institutions and health management departments to further promotion of the prevention work.

## 5 Conclusion

The findings of this meta-analysis are as follows: age, diabetes mellitus, renal disease, history of catheter-associated infection, hypertension, dialysis duration, catheter site, catheter duration, number of catheterizations, catheter type, CD4$^+$ cell count, albumin level, C-reactive protein level, hemoglobin level, procalcitonin level, inadequate hand hygiene, and APACHE II scores were identified as risk factors for CRBSI in HD patients. In contrast, no significant association was found between CRBSI and gender, anemia, external hospital tubing, cholesterol level, and poor patient hygiene. Of the included studies, only seven were cohort studies, with the remainder being retrospective case-control studies. Future research should endeavor to include more high-quality prospective cohort studies, featuring more rigorous designs to enhance the reliability of the research findings.

## Supporting information

**S1 File.**
(DOCX)

## Acknowledgments

We would like to thank Shuting Ren and Caiyun Zhang for their assistance with this study.

## Author Contributions

**Conceptualization:** Lili Wang.

**Data curation:** Huajie Guo, Ling Zhang.

**Formal analysis:** Lili Wang.

**Methodology:** Ling Zhang.

**Project administration:** Hua He.

**Resources:** Huajie Guo, Hua He.

**Software:** Ling Zhang, Hua He.

**Supervision:** Lili Wang.

**Writing – original draft:** Huajie Guo, Ling Zhang.

**Writing – review & editing:** Huajie Guo, Lili Wang.

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
