## [Decision Letter · Decision Letter 0]

6 Sep 2023

PONE-D-23-23774Risk factors for catheter-associated bloodstream infection in hemodialysis patients: a meta-analysisPLOS ONE

Dear Dr. Wang,

Thank you for submitting your manuscript to PLOS ONE. After careful consideration, we feel that it has merit but does not fully meet PLOS ONE’s publication criteria as it currently stands. Therefore, we invite you to submit a revised version of the manuscript that addresses the points raised during the review process.

We look forward to receiving your revised manuscript.

Kind regards,

Ankur Shah

Academic Editor

PLOS ONE

5. Please include your tables as part of your main manuscript and remove the individual files. Please note that supplementary tables (should remain/ be uploaded) as separate "supporting information" files.

Reviewers' comments:

Reviewer's Responses to Questions

**Comments to the Author**

1. Is the manuscript technically sound, and do the data support the conclusions?

Reviewer #1: Yes

Reviewer #2: Partly

2. Has the statistical analysis been performed appropriately and rigorously? 

Reviewer #1: Yes

Reviewer #2: I Don't Know

3. Have the authors made all data underlying the findings in their manuscript fully available?

Reviewer #1: Yes

Reviewer #2: Yes

4. Is the manuscript presented in an intelligible fashion and written in standard English?

Reviewer #1: Yes

Reviewer #2: Yes

5. Review Comments to the Author

Reviewer #1: Wang et al present an easy to read and well written meta-analysis on the risk factors associated with catheter associated bloodstream infections in hemodialysis patients. The meta-analysis has been registered to PROSPERO, the statistical analyses appear technically sound and the data are clear. However, there are some points that I need to address or questions that I would like to ask:

1-Introduction – first sentence: „Chronic kidney disease (CKD) represents a significant health concern due to its potential to cause irreversible kidney damage and eventual renal failure.“ – this statement is incorrect, or at least incorrectly worded, as CKD IS already the irreversible kidney damage and not its cause. Please rephrase.

2-Please state the search terms that were used to screen databases.

3-Has AI been used to generate the manuscript or parts of it?

4-Line 288: please correct the typo “studiy”

5-35 of the 49 included studies were from China (see Table 1). I am not sure whether national data can be used to draw international conclusions because HD practices vary at the national level. For example, are there standardized catheterization procedures or HD protocols in China? Have you performed all analyses separately for Chinese and international data? If yes, this should be mentioned and the data should be provided in the supplementary section.

6-Figure 1 is not of good quality and should have higher resolution

7-Figure 2: Please change the name of the x-axis

8-In my opinion, the discussion section would benefit from a brief discussion of the relevance of the data obtained to clinical practice and prevention of CRBSI.

Reviewer #2: This is a metaanalysis of risk factors associated with catheter-associated blood stream infections. 49 studies were identified (not 50 as stated in section 2.2 of the article). There are various risk factors that have been identified in this metaanalysis some of which were pulled from relatively small study sizes which reduces the validity of the results but these limitations have been noted by the authors.

The authors have also noted other limitations in the study design which reduce the robustness of the research.

However, it is well written, and relatively easy to understand and if the reader bears the limitations of the study in mind, it could prompt further research in the area.

6. PLOS authors have the option to publish the peer review history of their article (what does this mean?). If published, this will include your full peer review and any attached files.

Reviewer #1: No

Reviewer #2: No

---

## [Author Response · Author response to Decision Letter 0]

15 Jan 2024

Dear journal editors and reviewers:

Thank you for your careful reading and kind reminder of my manuscript.I admire your rigorous logic and pragmatic academic spirit.Following is my responds to each point raised.

1. The manuscript meets PLOS ONE's style requirements.

2. The authors has upload the minimal anonymized data set as Supporting Information files 

3.The corresponding author’s ORCID iD is validated in Editorial Manager. 

4.Each figure include a separate caption in the manuscript.

5.Two tables has been included as part of the manuscript.

Reviewers' comments:

Reviewer #1: Wang et al present an easy to read and well written meta-analysis on the risk factors associated with catheter associated bloodstream infections in hemodialysis patients. The meta-analysis has been registered to PROSPERO, the statistical analyses appear technically sound and the data are clear. However, there are some points that I need to address or questions that I would like to ask:

1-Introduction – first sentence: „Chronic kidney disease (CKD) represents a significant health concern due to its potential to cause irreversible kidney damage and eventual renal failure.“ – this statement is incorrect, or at least incorrectly worded, as CKD IS already the irreversible kidney damage and not its cause. Please rephrase.

The relevant content has been rephased in the manuscrip.(Line 67 and line68)

2-Please state the search terms that were used to screen databases.

The search strategy and search terms for each database have been uploaded as an appendix of Search Strategy file.

3-Has AI been used to generate the manuscript or parts of it?

The authors declares that they have not used AI to generate the manuscript or parts of it.The relevant content has been mentioned in the manuscript.(Line 64)

4-Line 288: please correct the typo “studiy”

Two typos have been corrected.(Line297 and line344) Meanwhile, the author checked all the possible typos in the manuscript.

5-35 of the 49 included studies were from China (see Table 1). I am not sure whether national data can be used to draw international conclusions because HD practices vary at the national level. For example, are there standardized catheterization procedures or HD protocols in China? Have you performed all analyses separately for Chinese and international data? If yes, this should be mentioned and the data should be provided in the supplementary section.

(1)After the editor's guidance,the authors realized this limitation in the study which reduce the extrapolation of the research(“strengths and limitations”part).

(2)The authors has analyzed the differences in Chinese and international research(3.2.7part)

(3)China is striving to keep pace with international standards by taking a series of measures at the national level.

After preliminary research and pilot projects, in 2021, the National Health Commission of China(NHCC) for the first time identified "reducing the incidence of intravascular catheter-related bloodstream infections" as one of the top ten goals for improving medical quality and safety .(https://www.gov.cn/zhengce/zhengceku/2021-02/22/content_5588240.htm)

In March 2021,the NHCC issued guidelines for the prevention and control of vascular catheter-related infections(http://www.nhc.gov.cn/yzygj/s7659/202103/dad04cf7992e472d9de1fe6847797e49.shtml)

 In this context, on the basis of soliciting clinical frontline opinions, repeated discussions, and evidence-based approaches,the Hospital Management Research Institute of the National Health Commission of China organized experts to develop (2021) and revise(2023) the "Quality Control Toolkit for the Prevention of Intravascular Catheter Related Bloodstream Infection" , which has been widely used in clinical practice.

The toolkit focuses on high-risk factors and key links of bloodstream infection during the placement and maintenance of different types of catheters, including sterile operating procedures, selection of catheters and puncture sites, catheter fixation and maintenance, and necessity assessment of catheter retention.Meanwhile,it comprised related checklists, implemented by hospitals nationwide.

In November of the same year, the NHCC issued the Standard Operating Procedures for Blood Purification (2021 Edition) which includes standardized catheterization procedures for HD patients.(http://www.nhc.gov.cn/yzygj/s7659/202111/6e25b8260b214c55886d6f0512c1e53f.shtml)

6-Figure 1 is not of good quality and should have higher resolution

The new figure with higher resolution has replaced the previous one.

7-Figure 2: Please change the name of the x-axis

The name of the x-axis has been corrected in Figure 2.

8-In my opinion, the discussion section would benefit from a brief discussion of the relevance of the data obtained to clinical practice and prevention of CRBSI.

The discussion section provides a more in-depth analysis of the significance of the results of this study for clinical guidance and the prevention of CRBSI.(section 3.3 and other discussion parts of the article)

Reviewer #2: This is a metaanalysis of risk factors associated with catheter-associated blood stream infections. 49 studies were identified (not 50 as stated in section 2.2 of the article). There are various risk factors that have been identified in this metaanalysis some of which were pulled from relatively small study sizes which reduces the validity of the results but these limitations have been noted by the authors.

The authors have also noted other limitations in the study design which reduce the robustness of the research.

However, it is well written, and relatively easy to understand and if the reader bears the limitations of the study in mind, it could prompt further research in the area.

The number of studies which were identified has been corrected in section 2.2(Line31 and Line170).Meanwhile, the author checked all the possible data errors in the manuscript.

---

## [Decision Letter · Decision Letter 1]

19 Feb 2024

Risk factors for catheter-associated bloodstream infection in hemodialysis patients: a meta-analysis

PONE-D-23-23774R1

Dear Dr. Wang,

We’re pleased to inform you that your manuscript has been judged scientifically suitable for publication and will be formally accepted for publication once it meets all outstanding technical requirements.

Kind regards,

Ankur Shah

Academic Editor

PLOS ONE

Additional Editor Comments (optional):

The authors have addressed all reviewers comments. Thank you.

Reviewers' comments:

Reviewer's Responses to Questions

**Comments to the Author**

1. If the authors have adequately addressed your comments raised in a previous round of review and you feel that this manuscript is now acceptable for publication, you may indicate that here to bypass the “Comments to the Author” section, enter your conflict of interest statement in the “Confidential to Editor” section, and submit your "Accept" recommendation.

Reviewer #1: All comments have been addressed

Reviewer #2: All comments have been addressed

2. Is the manuscript technically sound, and do the data support the conclusions?

Reviewer #1: Yes

Reviewer #2: Yes

3. Has the statistical analysis been performed appropriately and rigorously? 

Reviewer #1: Yes

Reviewer #2: I Don't Know

4. Have the authors made all data underlying the findings in their manuscript fully available?

Reviewer #1: Yes

Reviewer #2: No

5. Is the manuscript presented in an intelligible fashion and written in standard English?

Reviewer #1: Yes

Reviewer #2: Yes

6. Review Comments to the Author

Reviewer #1: The authors have addressed all my concerns. This meta-analysis contributes to a better understanding of risk factors leading to catheter-associated bloodstream infections in hemodialysis patients.

Reviewer #2: The authors have addressed all previous comments. The limitations of the study have been stated by the authors but it could still prompt further research.

7. PLOS authors have the option to publish the peer review history of their article (what does this mean?). If published, this will include your full peer review and any attached files.

Reviewer #1: **Yes: **Daniela Gerges

Reviewer #2: **Yes: **Ngozi Virginia Aikpokpo

---

## [Editor Report · Acceptance letter]

8 Mar 2024

PONE-D-23-23774R1 

PLOS ONE

Dear Dr. Wang, 

I'm pleased to inform you that your manuscript has been deemed suitable for publication in PLOS ONE. Congratulations! Your manuscript is now being handed over to our production team.

Kind regards, 

on behalf of

Dr. Ankur Shah 

Academic Editor

PLOS ONE